# Algal and Cyanobacterial Lectins and Their Antimicrobial Properties

**DOI:** 10.3390/md19120687

**Published:** 2021-12-01

**Authors:** José Abel Fernández Romero, María Gabriela Paglini, Christine Priano, Adolfina Koroch, Yoel Rodríguez, James Sailer, Natalia Teleshova

**Affiliations:** 1Department of Science, City University of New York Borough of Manhattan Community College, New York, NY 10007, USA; cpriano@bmcc.cuny.edu (C.P.); akoroch@bmcc.cuny.edu (A.K.); 2Center for Biomedical Research, Population Council, New York, NY 10605, USA; jsailer@popcouncil.org (J.S.); nteleshova@popcouncil.org (N.T.); 3Facultad de Ciencias Médicas, Instituto de Virología “Dr. JM Vanella”, Universidad Nacional de Córdoba, Córdoba X5000GYA, Argentina; gpaglini@immf.uncor.edu; 4Instituto de Investigación Médica Mercedes y Martín Ferreyra, INIMEC-CONICET-Universidad Nacional de Córdoba, Córdoba X5016GCT, Argentina; 5Department of Natural Sciences, City University of New York Hostos Community College, New York, NY 10451, USA; YRODRIGUEZ@hostos.cuny.edu

**Keywords:** lectin, antiviral, antibacterial, antiprotozoal, algal, cyanobacteria

## Abstract

Lectins are proteins with a remarkably high affinity and specificity for carbohydrates. Many organisms naturally produce them, including animals, plants, fungi, protists, bacteria, archaea, and viruses. The present report focuses on lectins produced by marine or freshwater organisms, in particular algae and cyanobacteria. We explore their structure, function, classification, and antimicrobial properties. Furthermore, we look at the expression of lectins in heterologous systems and the current research on the preclinical and clinical evaluation of these fascinating molecules. The further development of these molecules might positively impact human health, particularly the prevention or treatment of diseases caused by pathogens such as human immunodeficiency virus, influenza, and severe acute respiratory coronaviruses, among others.

## 1. Introduction

### 1.1. Lectins, Their Structure, Function, and Carbohydrate-Binding Specificity

The Latin root for “lectin” means to choose or select, an appropriate meaning given that lectins are proteins that “choose”: to bind carbohydrates in glycolipids or glycoproteins and that the interaction of lectins with carbohydrates can be very selective and as specific as the antigen/antibody interactions. In addition to binding to oligosaccharides, they might also bind to monosaccharides, although with less affinity. Lectins are ubiquitous and can be produced by different organisms, including animals, plants, fungi, protists, and microorganisms such as bacteria, archaea, or viruses. There are different types of lectins shown in Table 1 that have been classified based on structural and functional similarities. The different types of lectins include C-type lectins (selectins, collectins, and endocytic lectins), S-type lectins (galectins), siglecs (sialic acid-binding Ig-like lectins), L-type lectins, P-type lectins, M-type lectins, Jacalin-related lectin (JRL), Cyanovirin-N homologs (CVNHs), *Oscillatoria agardhii* agglutinin homolog (OAAH), and *Galanthus nivalis* agglutinin-like (GNA-like) lectins, among others [1].

Each lectin molecule usually contains several binding sites for the simultaneous binding to multiple units of the carbohydrates they target. The three-dimensional structures of these proteins and their binding to carbohydrates can be very different among various lectins. For example, griffithsin (GRFT; Table 1) comprises 121 amino acids, where residue 31 does not appear to correspond to any standard amino acid [2]. GRFT adopts a β-prism I motif [3] observed in a variety of lectins and other proteins [4]. This motif consists of three repeats of a four-stranded antiparallel β-sheet forming a triangular prism [5]. Different from other related lectins, GRFT is defined by a domain-swapped homodimer where the first 2 β-strands of one monomer are linked to 10 β-strands of the other monomer, and vice versa [6]. Several X-ray crystal structures of GRFT in complex with monosaccharides and disaccharides have been solved. These include mannose (PDB IDs 2GUC, 2GUD, and 3LL2), N-acetylglucosamine (PDB ID 2GUE), 1→ 6α−mannobiose (PDB ID 2HYQ), and maltose (PDB ID 2HYR) [6,7,8]. For example, Figure 1 shows the complex between GRFT and six mannoses corresponding to six carbohydrate (mannose)-binding sites. Each GRFT mannose-binding site encloses Tyr and Asp residues in which both amino acids form three hydrogen bonds with each mannose. The three GRFT mannose-binding sites in each monomer are arranged to create an almost perfect equilateral triangle.

Similarly, cyanovirin-N (CV-N; Table 1) is an elongated lectin and consists of two similar domains, each with five β-strands and two helical turns. These two domains are formed after domain swapping and are connected by helical turns [5]. This domain swapping is present in both CV-N and GRFT; however, CV-N dimers swap half of the molecule, while in GRFT only 2 β-strands out of 12 are swapped. Additionally, CV-N has been naturally isolated in monomeric and dimeric forms; whereas other lectins, such as *Microcystis viridis* lectin (MVL; Table 1) and GRFT, are naturally produced only in their dimeric form. Furthermore, other lectins such as scytovirin (SVN; Table 1) seem to be produced in the monomeric form [5].

The role lectins play varies in the different organisms that produce them; they are involved in various biological processes. For example, in mammalians, lectins can be involved in cell-to-cell self-recognition, gamete fertilization, embryonic development, cell differentiation, apoptosis, immunomodulation, and inflammation, among other functions. In the case of marine or freshwater organisms, the main topic of this review, the functions that lectins play have been associated with cell-to-cell recognition, attachment, and bioflocculation, providing a competitive advantage over other organisms in their natural habitat [9].

### 1.2. Algal and Cyanobacterial Lectins: Their Classification and Characteristics

Marine species account for about half of total biodiversity and thus have received significant attention for their ability to produce natural molecules with potential biomedical properties [10,11]. One type of these molecules comprises lectins, proteins with a high affinity for carbohydrates, particularly those found on the surface of many pathogens. Significant sources of biomedical lectins include cyanobacteria (17%), green algae (22%), and red algae (61%) [12].

Cyanobacteria are a diverse group of oxygenic photosynthetic prokaryotes commonly found in a broad range of aquatic and terrestrial environments. They flourish at the surfaces of lakes and oceans and form mats in benthic environments. They can tolerate higher temperatures than eukaryotic cells, and can tolerate high salinity environments, desiccation, and water stress. Cyanobacteria are also known as blue-green algae because they produce phycocyanin pigment, which gives the cells a bluish color when present in high concentrations [13]. In addition to playing a pivotal role in changing the composition of the planet’s atmosphere, several species have been studied for their ability to produce biomolecules of importance in the biomedical field.

Marine algae are diverse organisms: Rhodophyta (red algae) and Chlorophyta (green algae) differ in their pigments, anatomy, and reproduction. They are one of the oldest types of marine organisms on earth and require salty/brackish water and sunlight. They are usually found attached to rocky surfaces. Like cyanobacteria, marine algae constitute an essential source of biomolecules, particularly lectins, that have promising applications in biomedicine [14]. Table 1 summarizes some of the most relevant lectins identified in these marine species.

## 2. Antiviral Activity of Marine Lectins

### 2.1. Mannose Arrays on Viral Spikes as a Target for Lectins

Glycans (oligo- and polysaccharides) shield protein surfaces of viruses, providing protection against the enzymatic activity of proteases. A diverse number of oligosaccharides can be displayed on the surface of mammalian cells, which are not a target for lectins that generally require repetitive arrays of specific sugars for recognition. However, repetitive arrays, such as high-mannose arrays, can be found on the surface of pathogens. For this reason, and because this mannose content is uncommon in mammalian cells, lectins are selective antimicrobial agents capable of binding the pathogen’s glycoproteins with high affinity and are not likely to be toxic to human cells [15,52,53,54].

To understand better the role of these glycans and how they might constitute an important target for antimicrobial agents and immune response, it is necessary to take a closer look at glycosylation in the HIV spike protein. The HIV-1 viral spike is one of the most heavily glycosylated proteins. It is challenging for the host humoral response to produce anti-glycan antibodies. The glycan network on an HIV-1 spike is often referred to as the glycan shield. The main function of these glycans is to protect conserved regions of the spike protein from neutralizing antibodies. Still, some neutralizing antibodies capable of binding to the glycans on an HIV-1 spike can be generated by the host humoral response. Interestingly, a cluster of high-mannose glycans that includes the N-linked glycan at position 332 can be targeted by different broadly neutralizing antibodies (bNAbs) [55]. The main glycan-dependent sites on the HIV-1 spike include not only the N332 glycan/V3 loop in the intrinsic mannose patch, but also the N160 glycan/V1/V2 loops and the glycans near gp120/gp41 interface [56]. Some of the broadly neutralizing antibodies that interact with these glycans include PGT128, PGT121, 10-1074, PGT135, PG9, PG16, PGT145, CAP256-VRC26.25, CH04, PGT151, 35O22, and 8ANC195 [57,58,59,60,61,62,63,64,65]. As a result, bNAbs have appropriately been the focus of significant research in preventing, treating, or even curing HIV.

Lectins can play a similar role when compared to bNAbs, and in many cases have a broader spectrum of neutralization. GRFT, CV-N, and SVN can bind to these glycans with high affinity and neutralize virions at a half-maximal effective concentration (EC_50_) as low as 30 pM. These lectins are not HIV subtype-specific because similar high-mannose arrays are present in all HIV subtypes. The 234 and 295 glycosylation sites are involved in GRFT, CV-N, and SVN neutralization of HIV-1 [52]. HIV can develop resistance to lectins by deglycosylation in site-specific residues (glycans at positions 234 and 295); however, multiple mutations are necessary [66]. Consequently, the deglycosylation required for resistance development might render a virus more susceptible to neutralization by the immune system [67].

#### 2.1.1. Anti-HIV—Mode of Action and Efficacy Studies

HIV belongs to the viral family *Retroviridae*. It is an enveloped, positive-sense single-stranded RNA virus with the ability to reverse transcribe its genome and integrate the viral genome into the host genetic material. The viral reverse transcriptase controls the initial process of reverse transcription, and the lack of proofreading contributes to the high frequency of mutations that is clearly observed in the viral spike as a mechanism to evade humoral response. The initial process of viral replication starts with attachment to the CD4 receptor followed by interaction with a co-receptor, mainly CCR5 and CXCR4, and entry into a host cell. Many neutralizing molecules have been studied to block HIV entry into target cells. Because of the properties of lectins noted above, the potential for broad-spectrum efficacy, the high affinity, and the relatively low required concentration, lectins are increasingly being analyzed for their potential in this regard.

Recombinant GRFT has potent antiviral activity against HIV-1 strains and primary isolates with EC_50_ values in the low pM concentration. GRFT also interferes with cell-to-cell fusion and the transmission of HIV-1. GRFT blocks CD4-dependent glycoprotein (gp) 120 binding and inhibits gp120 binding of the monoclonal antibody (mAb) 2G12, which targets a carbohydrate-dependent motif, and the (mAb) 48d, that binds to CD4-induced epitope [2]. Mutations in the domain-swapped homodimer of GRFT, specifically in two or four amino acids at the dimerization interface, results in a monomeric form of GRFT. Cell-based assays indicate that this monomeric form of the lectin has greatly reduced activity against HIV-1, probably due to reduced crosslinking and aggregation of viral particles via multivalent interactions between GRFT and oligosaccharides present on HIV envelope glycoproteins [7]. Furthermore, GRFT, CV-N, and SVN are inhibitory for HIV-1 transfer from dendritic cells (DC-SIGN-mediated transfer of HIV-1) to CD4+ T cells, with GRFT being the most potent [68,69].

Different N-glycosylated sites have been implicated in mutations that affect the antiviral properties of lectins. Site-directed mutagenesis demonstrated that a single deglycosylation at N295 or N448 in HIV-1 isolates resulted in remarkable resistance to GRFT while still sensitive to CV-N, GNA, and some neutralizing antibodies. Unlike CV-N and GNA, the interaction between GRFT and gp120 appeared to be dependent on the specific trimer glycosylation, including N295 and N448 [67]. Deglycosylation of HIV-1 gp120 at position 295 or 448 decreases the sensitivity of transmitted/founder (T/F) gp120 to GRFT and at 339 to both CV-N and GNA. Mutation of all the three glycans renders a T/F gp120 resistant to GRFT [70]. HIV-1 subtype C develops resistance to GRFT, CV-N, and SVN after deletion of 1 to 5 mannose-rich glycans. Glycosylation sites at positions 230, 234, 241, and 289 located in the C2 region, and at positions 339, 392, and 448 in the C3-C4 region are involved in this resistance. The loss of glycosylation sites on gp120 and rearrangement of glycans in V4 are involved in HIV-1 subtype C resistance to GRFT, CV-N, and SVN [71].

MVN shares 33% identity with CV-N, and both lectins bind to similar carbohydrate structures. MVN and CV-N inhibit syncytium formation between HIV-1-infected T cells and uninfected CD4+ T cells and inhibit DC-SIGN-mediated HIV-1 binding and transmission to CD4+ T cells. HIV-1 exposure to dose-escalating concentrations of MVN selects a mutant virus with four deleted high-mannose-type glycans in gp120. The MVN-resistant virus is still sensitive to CV-N, HHA, GNA, and UDA but not to the 2G12 monoclonal antibody. Importantly, MVN is significantly less cytotoxic than CV-N [72].

Combinations of GRFT with tenofovir, maraviroc, and enfuvirtide showed synergistic activity against HIV-1 clade B and clade C isolates in primary peripheral blood mononuclear cells and in CD4+ MT-4 cells [20]. Similarly, paired carbohydrate-binding agent combinations show synergistic activity against HIV-1 and HIV-2, the 2G12 monoclonal antibody-resistant HIV-1, and MVN-resistant HIV-1. There is synergy in HHA and GNA combination against 2G12 monoclonal antibody-resistant HIV-1 and MVN-resistant HIV-1 strains. These carbohydrate-binding agents might have distinct binding patterns on the gp120 envelope and therefore might not interfere with each other’s glycan-binding sites on gp120 [73].

Few in vivo studies have been performed to test the efficacy of lectins against HIV-1. CV-N is effective against vaginal and rectal SHIV89.6P transmission [14]. Repeated administration of *L. jensenii* strain bioengineered to produce the protein CV-N in macaques led to a 63% reduction in transmission of SHIVSF162P3 after repeated challenge and in the peak viral loads in colonized macaques with breakthrough infection [14]. In the case of GRFT, only one study has shown the efficacy of this lectin against vaginal challenge in a macaque model. GRFT combined with carrageenan (CG) in a freeze-dried fast-dissolving insert (FDI) formulation protects rhesus macaques from a high dose vaginal SHIV SF162P3 challenge four hours after FDI insertion. The same formulation protects mice vaginally against HSV-2 and HPV pseudovirus [18].

#### 2.1.2. Anti-HSV—Mode of Action and Efficacy Studies

The *Herpesviridae* family includes important pathogens associated with human diseases. This is the case of HSV-1 and -2 enveloped DNA viruses. The entry of HSV into the host cell involves several viral glycoproteins displayed on the virion’s surface. The HSV glycoprotein C (gC) and glycoprotein B (gB) bind to heparan sulfate on the cell surface to start the process of viral attachment to host cells. Next, glycoprotein D (gD) binds to entry receptors leading to subsequent steps with the participation of other viral glycoproteins such as glycoproteins H (gH) and L (gL). Ultimately, the virus fuses with the host cell membrane and enters the cell to replicate and produce new viral progeny [74].

The participation of viral glycoproteins, highly glycosylated, is fundamental for the successful attachment and entry to target cells. These glycoproteins have been described as potential targets for lectins, such as GRFT, that can interfere with viral entry. Levendosky and collaborators showed that GRFT might inhibit entry but not attachment to target cells [24]. Glycoprotein gD seems to be pivotal in explaining the mode of action of GRFT as this glycoprotein participates in both viral entry and cell-to-cell spread but not in viral adsorption. Moreover, gD N-glycosylation sites, potential targets for GRFT, are close to the gD functional domains for binding the gD receptor [75]. Additionally, Nixon and collaborators have proposed that GRFT might inhibit HSV-2 replication by interfering with cell-to-cell spread and by primarily targeting gB [29]. In this study, Nixon and collaborators showed that a 0.1% GRFT gel, but not placebo gel, significantly reduced HSV-2 disease scores in a murine model and resulted in greater survival even in the presence of a human seminal plasma [29]. GRFT combined with CG in a gel reduced HSV-2 vaginal infection in mice when given one hour before challenge with a high HSV-2 dose (10^6^ plaque-forming units per mouse) [24]. Similarly, a GRFT and CG combination in a fast-dissolving insert (FDI) protected mice from HSV-2 infection four hours after FDI vaginal insertion [18].

#### 2.1.3. Anti-HCV—Mode of Action and Efficacy Studies

Hepatitis C virus (HCV) is a member of the *Flaviviridae* family that causes acute and chronic liver inflammation, sometimes leading to serious liver damage and infecting humans and chimpanzees through direct blood contact. HCV is a positive-sense, single-stranded enveloped RNA virus of approximately 9600 nucleotides in length, depending on the genotype. There are seven main HCV genotypes (numbered 1–7) that vary by over 30% in nucleotide sequence [76,77,78]. The genome of HCV encodes for a single polyprotein precursor of about 3000 amino acids that is cleaved by both viral and cellular proteases to generate 10 mature viral proteins. These include four structural proteins: C (Core), E1 and E2 (envelope proteins 1 and 2) and p7 (which functions as a proton channel); and six non-structural proteins (NS2, NS3, NS4A, NS4B, NS5A, and NS5B) that have enzymatic and other activities necessary for virus replication [79,80,81,82]. E1 and E2 are the two envelope glycoproteins present on the surface of viral particles and essential for virus attachment to its receptor/co-receptors and for subsequent internalization into the target cells via a pH-dependent and clathrin-mediated endocytosis [83,84,85]. Each protein comprises a large N-terminal ectodomain responsible for virus attachment to its receptors, and a C-terminal transmembrane domain that anchors each glycoprotein in the lipid bilayer. The N-terminal ectodomains are strongly glycosylated, and the structure is stabilized by disulfide bridges [82,86]. Interestingly, one-third of the total molecular mass of the heterodimers E1-E2 corresponds to N-glycans that are highly conserved from most genotypes. More-over, these sites harbor high-mannose-type glycans, even after egress of viral particles from the cells [87]. These features, and the conserved nature of the entry steps of HCV, offer an attractive target for using lectins to develop strategies to prevent infection, especially in the context of liver transplants and for therapeutic intervention. 

The progress in the development of retroviral particles pseudotyped with HCV envelope glycoproteins (HCVpp) and the amplification of HCV in cell culture (HCVcc) allow the identification and characterization of potential molecules that inhibit HCV entry to the host cell. In addition, it has been reported that the cyanobacterial lectins, CV-N, and MVL inhibit HCV entry by blocking the interaction between E2 and the receptor CD81 present in the membrane of the host cell. CV-N binds to high-mannose glycans (Man-8 and Man-9) in the E2 protein that forms part of the viral particle, exerting its inhibitory action at nanomolar concentrations in a dose-dependent manner. This was tested with different HCV genotypes in vitro using Huh-7 human hepatoma cells [11,88]. CV-N was found to be a more potent inhibitor than MVL, with EC_50_ concentrations ranging from 0.6 to 1.9 nM; whereas EC_50_ values of MVL ranged from 14.1 to 34.3 nM, more than 10-fold higher [89]. Another high-mannose-specific lectin, SVN, has also been shown to prevent HCV viral entry. In an infectivity assay using HCVpp bearing HCV E1 and E2 from various genotypes, SVN displayed inhibitory action with an EC_50_ of about 17 nM [32].

GRFT has been explored as a promising therapeutic antiviral strategy. GRFT exhibits a potent and broad-spectrum antiviral activity, with EC_50_ in the picomolar range. More-over, this lectin displays low in vitro and in vivo host toxicity and favorable subclinical outcomes range [90,91]. GRFT efficiently prevents HCV infection in cell culture (EC_50_ of 13.9 nM and EC_90_ of 49.5 nM), regardless of genotype, and it has been demonstrated that it also interferes with the direct cell-to-cell transmission of the virus (EC_50_ of 770 nM) [25]. The antiviral activity of GRFT lies in the ability to interact with the glycans present on the viral envelope proteins, E1-E2 heterodimer, thereby preventing the attachment of the virus to its receptor CD81 [25]. GRFT was successfully validated under in vivo conditions in chimeric mice that harbor primary human hepatocytes in their livers. Daily subcutaneous injections with 5 mg/kg GRFT could mitigate HCV infections. It is important to point out that GRFT can be safe and active when given on the same day as an HCV challenge [32]. The toxicity of GRTF in healthy mice was also evaluated, showing that treatments up to 50 mg/kg are well tolerated, with only mild toxicities [25,32].

The antiviral activity of all these lectins from red and blue-green algae against all genotypes of HCV can be explained by their targeting of high-mannose oligosaccharides on either E1 or E2 envelope proteins, which are common and conserved in all HCV genotypes.

#### 2.1.4. Anti-Influenza Virus—Mode of Action and Efficacy Studies

Influenza viruses are enveloped, negative-sense single-stranded, segmented RNA viruses of the Orthomyxoviridae family that are classified into A, B, C, and D types based on antigen differences in their matrix and nucleoprotein. Human influenza A and B viruses are highly contagious and are primarily responsible for the seasonally acute respiratory disease that outbreaks and spreads worldwide, giving rise to increased hospital admission and mortality. However, attention is focused mainly on influenza type A because it is by far the most virulent and causes severe respiratory disease or death [92,93].

Each virion contains three main components: a viral envelope of a lipid bilayer is decorated with three transmembrane proteins, namely hemagglutinin (HA), neuraminidase (NA), and an ion channel (M2); an intermediate layer of matrix protein (M1); and the third innermost component, the helical viral ribonucleocapsid (vRNP) core formed by the nucleoprotein (NP) and the eight segments of single-stranded negative-sense RNA (vRNA) [94]. The surface of the influenza A virus is covered with a large number of glycoproteins HA and NA, which are the basis of further subdivisions of the virus. To date, 18 subtypes of HA and 11 subtypes of NA have been identified by The United States Centers for Disease Control and Prevention (CDC); however, only H1-3 and N1-2 strains have achieved human-to-human transmission and are currently circulating as seasonal influenza strains (https://www.cdc.gov/flu/about/viruses/types.htm, accessed on 26 November 2021).

HA represents 80% of the surface viral glycoproteins in a ratio HA/NA of about 300/40, influenced by the genetic background and subtype of the virus [95]. These two glycoproteins, HA and NA, recognize the same molecule located at the host cell membrane’s surface, the N-acetylneuraminic (sialic) acid (SA) receptor [96]. HA plays two crucial roles during infection. First, it binds to the terminal SA on oligosaccharide chains of glycoproteins or glycolipids present at the host cell membrane, the first stage of viral infection. Further, it allows the fusion of viral and endosomal membranes, thereby delivering the viral nucleocapsid into the cytoplasm to initiate viral replication [97]. HA is a trimer of indistinguishable subunits, each one containing a receptor-binding site highly conserved in all subtypes of influenza [96]. NA is a tetramer of four identical polypeptide chains. This protein is a glycoside hydrolase mediating hydrolysis of the link between the SA bound to HA, relevant in promoting the release of new virus and preventing their aggregation on the host cell surface [98]. Thus, HA and NA have opposing roles, and the balance between their activities can be crucial in securing the infectivity of a particular viral strain. The functional and crucial role of HA in viral attachment and membrane fusion makes it an interesting target for antiviral drugs. In this regard, HA carries potential sites for N-linked glycosylation, is highly conserved between the different subtypes, and contributes to the stability of the molecule [99]. Hence, agents that interact with the glycans of HA to inhibit entry of the virus into susceptible target cells are excellent candidates.

Among potential agents, therefore, lectins that specifically recognize carbohydrate (glycan) structures are especially attractive for the development of prophylactic or therapeutic treatments of influenza virus infection. CV-N is highly active against a wide spectrum of influenza A and B virus strains, including clinical isolates and an in vitro-derived neuraminidase inhibitor-resistant strain. It exhibits antiviral activity by binding to high-mannose (oligomannose-8 and oligomannose-9) residues on HA1 molecules, resulting in virus neutralization with EC_50_ values from 0.004 to 0.5 g/mL, as tested in MDCK cells. However, there are two strains of the influenza virus, A/PR/8/34 (H1N1) and NWS/33 (H1N1), that are insensitive to CV-N even at concentrations of 10 g/mL. Loss of glycosylation sites of HA due to mutations leads to decreases in CV-N binding and antiviral efficacy [39,40,42]. The activity of CV-N was also assayed in animal models. In mice and ferrets, CV-N efficacy was dose-responsive from 0.0625 to 1 mg/kg/day when administered intranasally twice daily before viral infection. A significant reduction in viral titers in the lungs of mice and nasal washes in ferrets were observed [41]. However, CV-N has drawbacks in pharmaceutical applications, such as a short plasma half-life, proteolysis, and immunogenicity. PEGylation of CV-N as well as the linker-CV-N (LCVN) improved its pharmacokinetic and pharmacodynamic properties and maintained high affinity toward glycans containing mannose linkages [100]. These compounds, PEG_20k_-LCVN and LCVN, exhibited potent and selective anti-influenza activity in nanomolar concentrations in infected MDCK cells. In chicken embryos, they completely inhibited H3N2 virus propagation (strain A/HK/8/68) in micromolar concentrations [40,100].

The lectin BCA showed potent anti-influenza activity against most influenza virus strains in MDCK cells, including a clinical isolate of pandemic H1N1-2009 virus; however, the sensitivity was much lower (EC_50_ of 800 nM) compared with other strains [12,49].

Another two high-mannose-binding lectins, KAA-2 and HRL40, showed strong antiviral activity against influenza virus in MDCK and NCI-H292 cells, respectively. The activity was regardless of the virus strain and subtype, with EC_50_ in the low nanomolar levels. KAA-2 bound exclusively to high-mannose-type N-glycans, present at the globular head domain of HA, inhibiting the entry of the virus into the cell [12,47]. Similarly, HRL40 showed a potent anti-influenza activity through high-affinity binding to the viral hemagglutinin spike [50].

#### 2.1.5. Anti-EBOV—Mode of Action and Efficacy Studies

EBOV is the most prominent member of the *Filoviridae* family that includes enveloped viruses with a single-stranded, negative-sense RNA genome. Ebola virus disease (EBOVD) is a rare and deadly disease caused by infection with the EBOV. EBOV causes severe hemorrhagic fever in humans and nonhuman primates, with human case fatality rates of up to 90%. The virus first spreads to people through direct contact with the blood, body fluids, and tissues of animals. EBOV then spreads to other people through direct contact with the body fluids of a sick person or someone who has died from EBOVD [101,102].

The EBOV genome is a linear, non-segmented, single-stranded RNA of approximately 19 kb. The viral genome encodes seven proteins, including a nucleoprotein (NP), a glycoprotein (GP), an RNA-dependent RNA polymerase (L), and four structural proteins termed VP24, VP30, VP35, and VP40. Structural proteins VP40 and VP24 conform to the viral matrix linking the nucleocapsid to the viral envelope. GP is an integral membrane protein of the viral envelope that forms spike-like protrusions on the surface of the virion and is involved in the internalization process to the host cell [103,104]. GP is cleaved by furin into disulfide-linked GP1 and GP2 subunits [105,106]. GP1 is responsible for cellular attachment, while GP2 mediates viral and host membranes fusion [107]. The GP1 subunit contains the putative receptor-binding region and a heavily glycosylated region named mucin-like domain. GP1/2 molecule is a chalice-shaped trimer with a thick coating of oligosaccharides on a novel glycan cap and projecting mucin-like domain. The glycan cap and the mucin-like domain concentrate most of the N-linked glycosylation sites (including high-mannose oligosaccharides) and O-linked carbohydrates [107,108]. Thus, EBOV-GP is the main target for entry inhibitors, where lectins are excellent candidates. CV-N and SVN bind with high affinity to mannose-rich oligosaccharides on GP, blocking entry into target cells [51]. Antiviral activity of the lectins results from specific binding to the mucin region of GP, enriched in N-linked high-mannose oligosaccharides [109]. CV-N and SVN inhibit Zaire Ebola virus (EBOZV) infection, both in vitro and in vivo, through their ability to bind to oligomannoses-8/9 on the GP [36,51]. The use of recombinant systems to generate pseudotyped virus vectors carrying native GP or/and eGFP molecules has provided a powerful tool for studying the antiviral activity of several compounds. CV-N and SVN inhibit the replication of EBOZV-eGFP in Vero E6 cells with an EC_50_ of 154 nM and 41 nM [51]. Similar experiments using HeLa cells show that CV-N inhibits EBOZV GP-pseudotyped at lower concentrations (EC_50_ ~ 40–60 nM) [36]. On the other hand, when CV-N is used in vivo to treat EBOZV infected mice, repeated daily subcutaneous injections of approximately 5 mg/kg reduced the viral titers, and death was delayed but not prevented [37]. In vivo studies using SVN concluded that 90% of the mice receiving 30 mg/kg/day of SVN beginning on the day before virus inoculation survived the infection. Lower doses of SVN of 20 or 10 mg/kg/day showed 80% and 30% survival rates, respectively. Although SVN was highly protective in ZEBOV-infected mice, its short serum persistence required dosing every six hours [51], which is a disadvantage for any potential human therapy.

#### 2.1.6. Anti-Coronaviruses—Mode of Action and Efficacy Studies

The *Coronaviridae* family includes notable zoonotic pathogens, including MERS, SARS-CoV-1, and SARS-CoV-2. These viruses are enveloped viruses with a positive-strand RNA genome and highly N-glycosylated spike proteins that mediate attachment and entry to the epithelial cells that line the respiratory tract in humans. Different lectins have shown the potential to target coronaviruses [110]. O’Keefe and collaborators first reported that GRFT binds to the SARS-CoV-1 spike glycoprotein and inhibits viral entry. They showed the activity of GRFT against a variety of SARS-CoV-1 strains, other human coronaviruses, and coronaviruses that infect other animal species. Mice treated with GRFT and exposed to a mouse-adapted SARS-CoV-1 showed significantly reduced morbidity and mortality and inhibited cytokines associated with deleterious aspects of the host immunological response to the infection [30].

The combination of GRFT with ι-CG or λ-CG results in synergistic antiviral activity using a pseudoviral model. While ι-CG or λ-CG alone inhibits the entry of SARS-CoV-1 and SARS-CoV-2 pseudoviruses with EC_50_ values between 3.2 and 7.5 µg/mL, combination with GRFT results in 10 times lower EC_50_ values. This synergistic activity was observed with different pseudoviral particles, including those containing recently identified mutations in the SARS-CoV-2 spike protein (D614G, K1417N/E484K/N501Y) [16]. GRFT has also shown a synergistic effect when combined with a pan-coronavirus fusion inhibitor (EK1) that targets the SARS-CoV-2 spike S2 subunit [111]. Using particles pseudotyped with the MERS-CoV spike protein, Millet and collaborators showed that GRFT interferes with MERS-CoV entry [112].

#### 2.1.7. Anti-HPV—Mode of Action and Efficacy Studies

HPVs are small double-stranded DNA viruses in the *Papovaviridae* family with tropism for the skin and mucosal epithelia. Different from the other viruses discussed above, HPV is a naked virus, and the viral spikes are not glycosylated. Obviously, the virion per se cannot be targeted by lectins, but a unique mode of action might allow GRFT to interfere with HPV replication. Levendosky and collaborators proposed that GRFT might promote the internalization of the HPV secondary receptor in vitro. Additionally, the combination of GRFT and CG results in a potent anti-HPV activity in a murine model using HPV-16 pseudoviruses [18,24].

## 3. Antibacterial Activity of Marine and Freshwater Lectins

Though there are several examples in the literature of algal lectins as antivirals and several reports of lectins from many different sources that have antibacterial activity, there are relatively few studies in the literature that specifically address the potential usefulness of algal or cyanobacterial lectins as antibacterial agents [40,113,114]. This section will review the relevant information available. In all cases, standard microbiological methods were employed by researchers to test for inhibition of bacterial growth, including disk diffusion and culture density assays.

In one study, Liao et al. [115] showed that purified lectins isolated from two red algal species, *Eucheuma serra* (ESA) and *Galaxaura marginate* (GMA), strongly inhibited the growth of the pathogenic marine Gram-negative *Vibrio vulnificus*, although it showed no activity against two other Vibrio species, *V. peagius* and *V. neresis*. Selective inhibition was attributed to differences in bacterial surface carbohydrates. In addition, this study showed that whereas saline and ethanol extracts of several algal species exhibited antibacterial activity against both *V. vulnicius* and *V. peaguius*, this activity was inhibited by pre-treatment with lectin-binding sugars and glycoproteins, suggesting that lectins present in the algal extracts were the active inhibitory agents. Hung et al. [116] further demonstrated that lectins isolated from the red algae *Eucheuma denitculatum* (EDA) exhibit activity against another pathogenic marine vibrio, *V. alginolyticus*, but not against *V. parahaemolyticus* or *V. harveyi*. Binding assays suggested that selective activity was through binding of the EDA lectins to high-mannose N-glycans.

Holanda and colleagues reported differential activity of lectins from the marine red alga *S. filiformis* against human pathogenic bacteria [117]. In this study, isolated lectins inhibited the growth of Gram-negative bacteria *Salmonella typhi*, *Serratia marcescens*, *Enterobacter aerogenes*, *Klebsiella pneumoniae*, *Pseudomonas aeruginosa*, and *Proteus* sp. There was no inhibitory effect, however, on Gram-negative *S. typhimurium* or *Escherichia coli,* nor was there any inhibition of Gram-positive *Staphylococcus aureus*, *Bacillus subtilis,* or *Bacillus cereus*. For inhibited strains, bacterial cell density was significantly reduced in the presence of lectin at high concentrations (1 mg/mL), but cell density assays showed that bacteria exhibited all phases of cell growth. The investigators suggested that mannan present on cell walls of Gram-negative bacteria might bind to lectin and alter the flow of nutrients, thereby causing inhibition of growth [45,113].

Vasconcelos et al. tested the effect of isolated lectins from two species of red algae, *Bryothamnion seaforthii* (BSL) and *Hypnea musciformis* (HML), for their ability to inhibit growth in biofilms of Gram-positive *S. epidermidis* and *S. aureus*, and of Gram-negative *Klebsiella oxytoca*, *P. aeruginosa*, *Candida albicans*, and *Candida tropicalis*. Whereas HML and BSL both caused weak growth reductions in *S. aureus*, *S. epidermidis,* and *P. aeruginosa*, only HML reduced the growth of *K. oxytoca* [118]. For *S. aureus*, BSL decreased the biofilm mass at all concentrations used; however, HML caused only a small decrease at the highest concentration tested (250 µg/mL). In addition, these lectins caused only a small decrease in the number of viable *S. aureus* cells. The biofilm mass of *K. oxytoca* was also reduced in the presence of these lectins, but with no decrease in the number of viable cells [118].

Collectively, these investigators have suggested the potential usefulness of marine lectins as natural alternatives to conventional antibiotics in therapeutic interventions for infections by Gram-negative pathogens [117] and for the protection of marine species susceptible to marine vibrio infection [40,113,114]. With the global rise in antibiotic resistance over the past century, there has been a pressing need to find alternative sources of antibiotic agents for therapeutic use. As demonstrated in the studies reviewed here, the ability of algal lectins to inhibit the growth of various pathogenic bacteria makes them potential candidates for medicinal use and is worth further investigation.

## 4. Antiprotozoal Activity of Marine Lectins

Although metabolite extracts obtained from some marine algae have been shown to exhibit antiprotozoal activity, there are few reports in the literature of activity specifically attributed to algal or cyanobacterial lectins. Reports include the activity of algal extracts from *Bostrychia tenella* against *Trypanosoma cruzi* and *Leishmania amazonensis* [119]; from various species of brown algae against *T. cruzi*, *Trypanosoma brucei rhodesiense*, *L. donovani*, *Sargassaceae* sp. [120]; and from *Anadyomene saldanhae*, *Caulerpa cupressoides*, *Canistrocarpus cervicornis*, *Dictyota* sp., *Ochtodes secundiramea*, and *Padina* sp. against *L. braziliensis*. Chatterjee et al. further report that GRFT exhibits anti-protozoal activity against *T. vaginalis* and *Tritichomonas foetus* in a vaginal mouse model [17]. This is the only report of an algal lectin having such activity. Investigation into the possible activity of other marine lectins might yield novel insights into therapeutic uses against parasitic protozoan infections.

## 5. Expression of Marine Lectins in Heterologous Systems

Expressing lectins in heterologous systems can lead to a cost-effective production for pharmaceutical purposes. The heterologous systems provide higher yields than conventional purification and reduce the production cost and time [121]. Several models have been used for the heterologous production of lectins, such as bacteria, yeast, plants, mammalian, and insect cells. Figure 2 shows different strategies used for the expression and purification of GRFT and their intended use.

GRFT is the marine lectin in which expression in heterologous systems has been more widely explored. The expression of GRFT in tobacco plants (*Nicotiana benthamiana*) has been one of the choices. For this purpose, a synthetic cDNA (GenBank no. FJ594069) encoding the 121 amino acids of GRFT has been cloned into a tobacco mosaic virus (TMV) vector in which GRFT is expressed under the control of a duplicated coat protein subgenomic promoter. *N. benthamiana* seedlings are inoculated with infectious recombinant TMV, and the infected leaf biomass is processed 12 days after infection to extract GRFT [31]. The studies have shown that GRFT becomes the most abundant protein in the plant material and can be purified through filtration and chromatography. Through this process, GRFT accumulates at more than 1 g of GRFT per kilogram of *N. benthamiana* leaf material, providing a final product at concentrations above 20 mg/mL, and allowing production of more than 60 g of pure GRFT in a single 5000-square-foot enclosed greenhouse. The manufacturing cost of GRFT using this plant-based system has been estimated to be USD $0.32/dose. This assumes a commercial launch volume of 20 kg GRFT/year for 6.7 million doses of GRFT at 3 mg/dose, a recovery efficiency of 70%, and purity of >99%. This manufacturing process was also found to have a favorable environmental output with minimal risks to health and safety [122]. Additionally, gene-silencing suppressors for high-level production of GRFT in *N. benthamiana* have resulted in a higher accumulation of GRFT with a yield of 400 μg g^−1^ fresh weight or 287 μg g^−1^ after purification, representing a recovery of 71.75% [123].

Similarly, *Nicotiana tabacum* has been transformed with a vector containing the gene encoding CV-N. Through this process, the plant-derived CV-N can be recovered at 130 ng per mg of fresh leaf tissue. CV-N is expressed in the desired monomeric form using this plant-based system. Hydroponic culturing of transgenic plants results in CV-N rhizosecretion at 0.64 mg/mL hydroponic media after 24 days [124]. The transplastomic plants allow a highly efficient and cost-effective production platform for lectins, and dried tobacco can serve as a source material for the purification of lectins [125].

*Oryza sativa* (rice plant) has been used to express GRFT. For this purpose, GRFT has been expressed in the endosperm of transgenic rice plants. The yield of GRFT in this system can reach 223 g/g dry seed weight, and through a one-step purification protocol, can achieve a recovery of 74% and a purity of 80% [126].

Other host organisms for the economical and efficient production of lectins include bacteria. Hexa-histidine-tagged GRFT (His-GRFT) has been successfully produced in *E. coli*. Production in a fermenter with an auto-inducing medium allows the total amount of His-GRFT per liter to be increased by about 45-fold [127]. Similarly, recombinant expression in engineered *E. coli* results in GRFT concentrations of 2.5 g/L. This could translate into production volumes of >20 tons per year at the cost of goods sold below USD $3500/kg [128].

Finally, probiotic lactobacilli have been studied for the expression of lectins. *Lactobacillus rhamnosus* GG and GR-1 have been engineered to express GRFT [129], while *Lactobacillus jensenii* and *Lactobacillus plantarum* have been used to express CV-N [130,131,132] and SVN [133], respectively. The idea behind this strategy is that lectin-producing lactobacilli could colonize the mucosal epithelium and produce the lectin in vivo to protect the host if exposed to HIV or other pathogens.

## 6. Preclinical and Clinical Safety Studies of Marine and Freshwater Lectins

In addition to the potent antimicrobial activity, the safety of potential lectin-based products is of paramount importance. Comprehensive preclinical and clinical safety evaluations must be performed as part of product development [134]. This section reviews the preclinical and clinical studies that have been performed for these lectins.

*CV-N:* CV-N has been of particular interest for the development of a topical anti-HIV microbicide. Prolonged production of recombinant CV-N by *L. jensenii* vaginally in nonhuman primates did not induce any observable adverse effects or inflammatory biomarkers [135]. However, there are some safety concerns over the use of CV-N. CV-N affected PBMCs morphology, induced mitogenic activity in PBMCs, and increased expression of cellular activation markers and several cytokines following prolonged exposure [136,137]. CV-N can bind to cellular proteins and might induce potential toxic effects [89]. Furthermore, some paradoxical effects of CV-N enhancing R5 HIV infection at low concentrations were reported [138]. CV-N has been modified by site-specific conjugation with polyethylene glycol in a reaction called PEGylation to improve the drug-like properties of this lectin [139]. When administered intravenously, the PEGylated CV-N was significantly less immunogenic than CV-N [139]. The safety concerns observed in vitro have reduced the enthusiasm for this lectin.

*MVN and MVL:* MVN isolated from *Microcystis aeruginosa* shares partial homology with CV-N, has a potent but narrow anti-HIV profile, and demonstrates a better safety profile than CV-N [72]. However, several cytokines were significantly increased in PBMCs after exposure to this lectin [72]. The incubation of PBMCs with MVN leads to weak induction of expression of activation markers but did not activate or enhance viral replication in pretreated cells [72,140].

Like CV-N, MVL binds to the target cell surface and the viral envelope [89]. Cytotoxic effects triggered by the lectin might occur because of MVL interaction with cellular proteins. Indeed, MVL inhibited cell viability in several cell lines, including Hep-G2 (human hepatocellular liver carcinoma), HT-29 (human colon cancer), SGC-7901 (stomach cancer), and SK-OV-3 (human ovarian cancer) (IC_50_ 40–53 µg/mL)) [141].

*OAA:* OAA is a stable protein [142]. However, the development of OAA-based products might be problematic because it exerts cytotoxic effects such as CV-N, MVN, and MVL [44].

*GRFT:* GRFT has an excellent safety profile. In contrast to CV-N, GRFT, with its broad and potent antiviral activity, does not have stimulatory properties [143]. The lectin inhibits HIV infection in human cervical explant tissues with no proinflammatory cytokine production. GRFT has an excellent safety profile when tested in a rabbit vaginal irritancy model [31] or when administered in single or chronic subcutaneous doses in mice and guinea pigs [144]. GRFT is safe and minimally absorbed after repeated vaginal application. Repeated dosing of GRFT and GRFT/CG gel in small animal models revealed no adverse findings at any dose levels tested and showed that a GRFT/CG gel is non-irritating. Seven days of daily vaginal application of 0.1% GRFT/CG gel did not enhance the susceptibility of mice to HSV-2 infection. Fourteen days of daily intravenous administration of GRFT up to 8.3 mg/kg/day in rats resulted in no detectable anti-drug-antibodies (ADA) and a no adverse effect level (NOAEL) of 8.3 mg/kg/day despite high systemic levels of GRFT. Fourteen days of daily vaginal GRFT/CG gel dosing (up to 0.3% GRFT) in rats resulted in a NOAEL of 0.3% GRFT and little or no vaginal irritation. This regimen also resulted in little or no systemic detection of GRFT. A related study in rabbits also found a NOAEL of 0.3% GRFT and little or no vaginal irritation [18].

GRFT’s preclinical safety, lack of systemic absorption after vaginal administration in animal studies, and lack of cross-resistance with existing antiretroviral drugs prompted its development for topical HIV pre-exposure prophylaxis (PrEP). The Population Council investigated the safety, pharmacokinetics, pharmacodynamics, and immunogenicity of a vaginal gel (PC-6500: 0.1% GRFT in a CG gel) in healthy women after vaginal administration. In this first-in-human trial of GRFT, no significant adverse events were recorded in clinical or laboratory results or histopathological evaluations in cervicovaginal mucosa. Additionally, no anti-drug (GRFT) antibodies were detected in serum, and no cervicovaginal proinflammatory responses or changes in the ectocervical transcriptome were evident. Decreased levels of proinflammatory chemokines in CVLs were observed, while GRFT was not detected in plasma after vaginal application. GRFT and GRFT/CG in CVL samples inhibited HIV and HPV, respectively, ex vivo in a dose-dependent manner. This study suggested that GRFT formulated in combination with CG is a safe and promising multipurpose prevention technology product that warrants further investigation (Teleshova et al. submitted). The Population Council is currently investigating fast-dissolving inserts and vaginal rings containing Q-GRFT alone or in combination with CG. Q-GRFT is a version of GRFT in which a methionine has been substituted by another amino acid (M78Q) to reduce the potential oxidation of this lectin. An additional phase 1 clinical trial (PREVENT), led by the University of Louisville and the University of Pittsburgh, was planned to look at a GRFT-based rectal microbicide safety. This trial was terminated, and the study enrollment was prematurely halted due to the COVID-19 pandemic. The trial was designed to study if a single dose of an enema containing Q-GRFT was safe, well tolerated, and acceptable in healthy adults practicing receptive anal intercourse [145].

Limited preclinical safety data are available for several promising lectins, including SVN, BCA, KAA-2, and HRL40.

## 7. Conclusions

The *sui generis* mode of action of mannose-binding lectins such as GRFT, SVN, CV-N, OAA, and MVL against important pathogens has prompted the development of these molecules as potential therapeutics or prophylactic drugs to target single or multiple infectious diseases. Lectins are highly specific, might show broad-spectrum activity, are locally (topically) delivered, and are relatively potent. As such, they deserve additional investigation. That said, there are safety concerns with some lectins that induce mitogenic activity after prolonged exposures, and mass production of lectins is a relatively nascent field. GRFT might be an especially promising candidate. GRFT has activity that seems to be among the broadest of any yet-evaluated lectin, is an exception to the mitogenic activity, has been tested in phase 1 clinical trials with promising results regarding its safety in topical formulations, and appears to be producible in cost-effective doses.

## Figures and Tables

**Figure 1 marinedrugs-19-00687-f001:**
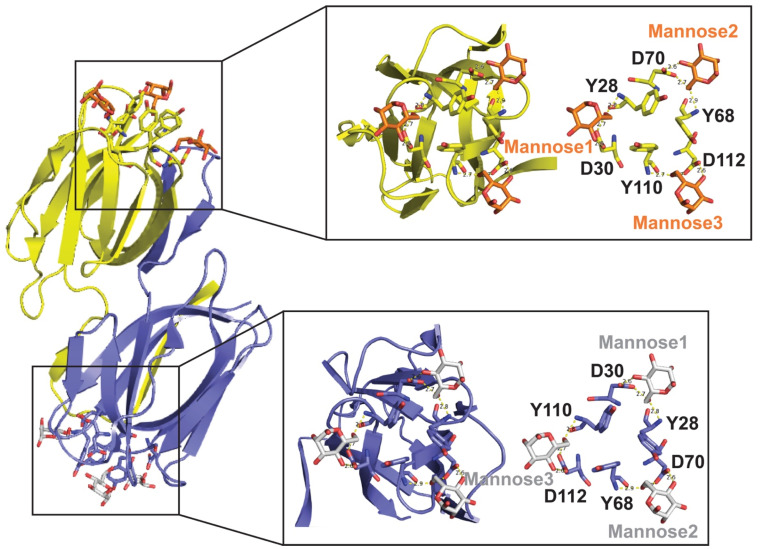
**Three-Dimensional Structure of Griffithsin (GRFT)-Mannose Complex (PDB ID 2GUD at 0.94 Å resolution). Left-Panel:** GRFT domain-swapped dimer. Six carbohydrate (mannose)-binding sites are shown; three for each monomer in yellow and purple, respectively. **Right-Panel:** Magnification of the GRFT six mannose-binding sites shown in the presence (left) and absence (right) of the β-strands cartoon representation. GRFT-Mannoses main interactions via hydrogen bonds are shown as yellow dashed lines. The three GRFT mannose-binding sites together form an almost perfect equilateral triangle.

**Figure 2 marinedrugs-19-00687-f002:**
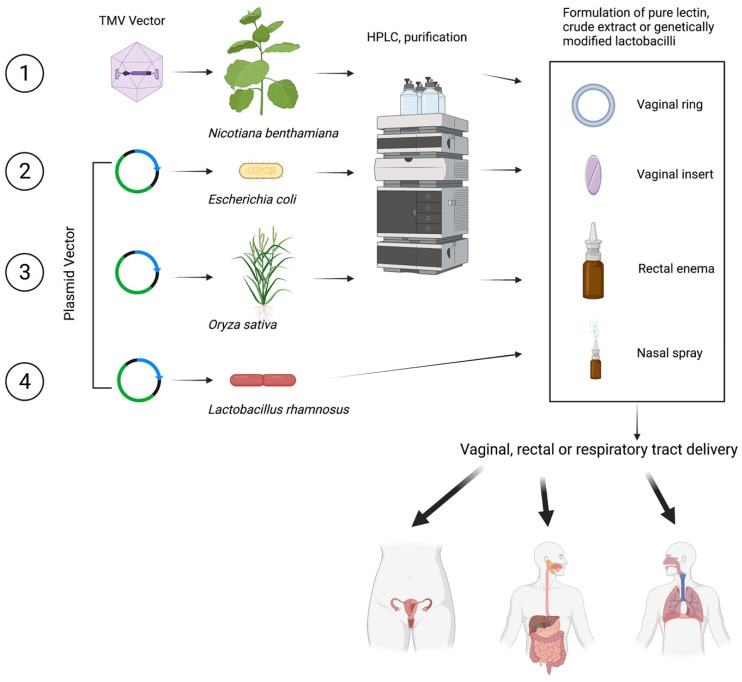
**Strategies to produce and purify GRFT in heterologous systems**. Created with BioRender.com.

**Table 1 marinedrugs-19-00687-t001:** Marine and freshwater lectins produced by algae and cyanobacteria.

Lectin	Lectin Family	Carbohydrate-Binding Specificity	Marine Species That Produce Each Lectin	Antimicrobial Properties	References
GRFT	JRL	High-mannose glycans HIV-1 gp120 K_D_ ^j^ 80 pM	Produced by *Griffithsia sp.* *Griffithsia* is a genus of red algae that includes about 27 species worldwide, most of which occur intertidally along temperate to tropical oceans in Europe, South America, and Africa.	HIV ^a^ HSV ^b^ HCV ^c^ SARS-CoV-1 and 2 ^d^ MERS CoV ^e^ EBOV ^f^ JEV ^g^ HPV ^h^ *Trichomonas vaginalis*	[2,7,15,16,17,18,19,20,21,22,23,24,25,26,27,28,29,30,31,32,33,34,35]
CV-N	CVNHs	High-mannose glycans HIV-1 gp120 K_D_ 6–45 nM	Produced by *Nostoc ellipsosporum. Nostoc* is a genus of oxygenic photosynthetic cyanobacteria that are widely distributed in terrestrial and aquatic habitats. *Nostoc* can resist desiccation, freezing, and thawing; and can fix atmospheric N_2_.	HIV HCV Influenza virus Rhinoviruses SARS-CoV-2 EBOV Measles virus HHV-6 ^i^ *Trichomonas vaginalis*	[17,36,37,38,39,40,41,42]
MVL	CVNHs	High-mannose glycans HIV-1 gp120 K_D_ 70–100 nM	MVL and MVN are produced by *Microcystis viridis and Microcystis aeruginosa,* respectively. *Microcystis* is a genus of freshwater cyanobacteria. They are planktonic and thrive in warm, stagnant waters worldwide.	HIV-1 HCV	[43]
Microvirin (MVN)	CVNHs	High-mannose glycans HIV-1 gp120 K_D_			
OAA	OAAH	High-mannose glycans HIV-1 gp120 K_D_ 0.6 nM	Produced by *Oscillatoria agardhii. Oscillatoria* is a genus of filamentous cyanobacteria. The genus name refers to the oscillation in the organism’s movement. They are found in freshwater.	HIV-1	[44]
SfL	OAAH	High-mannose glycans	Produced by *Solieria filiformis. S. filiformis* is a red alga found in subtropical warm waters along the coast in the North Atlantic Ocean, Gulf of Mexico, Colombia, Brazil, West Africa, Mediterranean, Israel, and Arabian Gulf.	*Serratia marcescens* *Salmonella typhi* *Klebsiella pneumoniae Enterobacter aerogenes Proteus* sp. *Pseudomonas aeruginosa*	[45]
KAA-2	OAAH	High-mannose glycans	Produced by *Kappaphycus alvarezii. K. alvarezii* is a red alga that occurs naturally in the south of the Philippines and is also cultivated in the South Pacific. *K. alvarezii* grows in a wide variety of environments.	HIV-1 Influenza virus	[40,46,47,48]
BCA	GNA-like	High-mannose glycans HIV-1 gp120 K_D_ 2.7 nM	Produced by *Boodlea coacta. B. coacta* is a green alga widespread throughout the tropics and can be seasonally dominant on some Indo-West Pacific reef-flats.	HIV-1 Influenza virus	[12,49]
HRL40	Unknown	High-mannose glycans	Produced by *Halimeda renschii. H. renschii* is a segmented-marine green alga that occurs naturally in the Indo-Pacific region and the Atlantic Ocean. It is a major contributor to marine sediments in the tropics and subtropics.	Influenza virus	[50]
SVN	Unknown	High-mannose glycans	Produced by *Scytonema varium. Scytonema* includes cyanobacteria generally considered to be cosmopolitan. *S. varium* is a freshwater or terrestrial species.	HIV HCV SARS-CoV-1 EBOV	[51]

^a^ HIV: human immunodeficiency virus; ^b^ HSV: herpes simplex virus; ^c^ HCV: hepatitis C virus; ^d^ SARS-CoV-1 and 2: severe respiratory syndrome coronavirus 1 and 2; ^e^ MERS-CoV: Middle East respiratory syndrome coronavirus; ^f^ EBOV: Ebola virus; ^g^ JEV: Japanese encephalitis virus; ^h^ HPV: human papillomavirus; ^I^ HHV-6: human herpes virus 6. ^j^ K_D_ = dissociation constant after binding to HIV-1 gp120.

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
