# Peer review of "Algal and Cyanobacterial Lectins and Their Antimicrobial Properties"

_marinedrugs, 2021, doi:10.3390/md19120687_

Round 1

Reviewer 1 Report

 In this review, authors mainly introduced the structure, function, classification, antimicrobial properties of algal and cyanobacterial Lectins. Their expression in heterologous systems and the preclinical and clinical evaluation were also analyzed. Generally, the review has covered most areas of lectin studies, and the writing is succinct in style. I have few questions as following:

Question1:

  1. Since most of the manuscript describes the antiviral activity of lectins, I suggest making a summary table to give readers a simple and direct introduction.
  2. The authors mainly focuse on the mode of action and efficacy of lectins. The innovation of this review should be further improved. Such as the medicinal application of lectins (See publication PMID: 25794876) and the biomedical perspectives of marine lectin used as blockers for coronaviruses (See publication PMID: 34203435).
  3. Comparing the carbohydrate-binding specificity of mannose-binding lectins from different species is helpful to deepen readers understanding of the properties of algal and cyanobacterial lectins.

Author Response

We thank the reviewers for their constructive criticisms and recognize they aim to improve the quality of the manuscript. Please see below our answer to each criticism.

 Reviewer 1

Comments and Suggestions for Authors

In this review, authors mainly introduced the structure, function, classification, antimicrobial properties of algal and cyanobacterial Lectins. Their expression in heterologous systems and the preclinical and clinical evaluation were also analyzed. Generally, the review has covered most areas of lectin studies, and the writing is succinct in style. I have few questions as following:

Question1:

  1. Since most of the manuscript describes the antiviral activity of lectins, I suggest making a summary table to give readers a simple and direct introduction.

Answer: Table#1, included in the introduction (section 1.2), has the list of lectins, and one column summarizes the antimicrobial properties (including the viruses they may target).

  1. The authors mainly focus on the mode of action and efficacy of lectins. The innovation of this review should be further improved. Such as the medicinal application of lectins (See publication PMID: 25794876) and the biomedical perspectives of marine lectin used as blockers for coronaviruses (See publication PMID: 34203435).

Answer: Both references have been now included in the manuscript. You can find in different sections of our manuscript information about potential medicinal application of lectins in terms of antimicrobial properties (that is the focus and title of our paper). Please notice the manuscript includes a section (section 2.1.6) that summarizes the potential use of lectins to treat or prevent coronavirus infections (including SARS-CoV-1 and 2).

  1. Comparing the carbohydrate-binding specificity of mannose-binding lectins from different species is helpful to deepen readers understanding of the properties of algal and cyanobacterial lectins.

Answer: Section 1.1 title has been updated to:

  • LECTINS, THEIR STRUCTURE, FUNCTION AND CARBOHYDRATE-BINDING SPECIFICTY.

The carbohydrate-binding specificity for each lectin is now included in Table 1.

Reviewer 2 Report

This review focuses on lectins produced by marine or freshwater organisms, in particular algae and cyanobacteria, and explore their structure, function, classification, and antimicrobial properties. It also describes the expression of lectins in heterologous systems and the current research on the preclinical and clinical evaluation of the molecules.

However, most of references cited in this review are rather old in spite of many relevant papers published recently. For example, a review published by Barre et al. covers the above contents with updated information.

Barre, A.; Van Damme, E.J.M.; Simplicien, M.; Le Poder, S.; Klonjkowski, B.; Benoist, H.; Peyrade, D.; Rougé, P.:  Man-Specific Lectins from Plants, Fungi, Algae and Cyanobacteria, as Potential Blockers for SARS-CoV, MERS-CoV and SARS-CoV-2 (COVID-19) Coronaviruses: Biomedical Perspectives. Cells 2021, 10, 1619. 

Author Response

We thank the reviewers for their constructive criticisms and recognize they aim to improve the quality of the manuscript. Please see below our answer to each criticism.

Reviewer 2

 Open Review

Comments and Suggestions for Authors

This review focuses on lectins produced by marine or freshwater organisms, in particular algae and cyanobacteria, and explore their structure, function, classification, and antimicrobial properties. It also describes the expression of lectins in heterologous systems and the current research on the preclinical and clinical evaluation of the molecules.

However, most of references cited in this review are rather old in spite of many relevant papers published recently. For example, a review published by Barre et al. covers the above contents with updated information.

Barre, A.; Van Damme, E.J.M.; Simplicien, M.; Le Poder, S.; Klonjkowski, B.; Benoist, H.; Peyrade, D.; Rougé, P.:  Man-Specific Lectins from Plants, Fungi, Algae and Cyanobacteria, as Potential Blockers for SARS-CoV, MERS-CoV and SARS-CoV-2 (COVID-19) Coronaviruses: Biomedical Perspectives. Cells 2021, 10, 1619.

Answer: The reference provided by the reviewer has been added. Please notice our manuscript is a general review that summarizes the antimicrobial properties against different pathogens, not only coronaviruses. We disagree with the statement that the references used in our manuscript are old. Please see below a comparison of the references used in our manuscript and the one mentioned by the reviewer (Barre et al. 2021). The table (see the attached PDF document) shows a breakdown by year. We are not using old references.

Finally, you stated that you are not qualified to judge about the English language and style. However, when rating if the English used was correct and readable you gave one star (that means very poor). I guess it was unintentional.

Round 2

Reviewer 2 Report

The merit of this review may be strengthen by focussing on GRFT.